# Update on Epidemiology, Diagnosis, and Biomarkers in Gastroenteropancreatic Neuroendocrine Neoplasms

**DOI:** 10.3390/cancers14051119

**Published:** 2022-02-22

**Authors:** Daisuke Takayanagi, Hourin Cho, Erika Machida, Atsushi Kawamura, Atsuo Takashima, Satoshi Wada, Takuya Tsunoda, Takashi Kohno, Kouya Shiraishi

**Affiliations:** 1Division of Genome Biology, National Cancer Center Research Institute, 5-1-1 Tsukiji, Chuo-ku, Tokyo 104-0045, Japan; dtakayan@ncc.go.jp (D.T.); hcho@ncc.go.jp (H.C.); ermachid@ncc.go.jp (E.M.); atkawamu@ncc.go.jp (A.K.); atakashi@ncc.go.jp (A.T.); tkkohno@ncc.go.jp (T.K.); 2Department of Clinical Diagnostic Oncology, Clinical Research Institute for Clinical Pharmacology and Therapeutics, Showa University, 6-11-11 Kitakarasuyama, Setagaya-ku, Tokyo 157-8577, Japan; st-wada@med.showa-u.ac.jp; 3Department of Medicine, Division of Medical Oncology, Showa University School of Medicine, 1-5-8 Hatanodai, Shinagawa-ku, Tokyo 142-8666, Japan; ttsunoda@med.showa-u.ac.jp; 4Department of Surgery, Saitama Medical Center, Jichi Medical University, 1-847 Amanuma-cho, Omiya-ku, Saitama 330-8503, Japan; 5Division of Gastroenterology and Hepatology, Department of Internal Medicine, The Jikei University School of Medicine, 3-25-8, Nishi-shimbashi, Minato-ku, Tokyo 105-8461, Japan; 6Department of Gastrointestinal Medical Oncology, National Cancer Center Hospital, Tsukiji, Chuo-ku, Tokyo 104-0045, Japan

**Keywords:** gastroenteropancreatic neuroendocrine tumor, carcinoid tumor, genetics, biomarkers

## Abstract

**Simple Summary:**

Neuroendocrine neoplasms are divided into two groups: well-differentiated neuroendocrine tumors and poorly differentiated neuroendocrine carcinomas. The progress in diagnostic methods, including pathology optimization and imaging, might be one of the reasons for the increasing incidence of gastroenteropancreatic neuroendocrine neoplasms; however, the remaining biological factors are undetermined. Rapid advances in molecular diagnostic and treatment strategies in recent years have significantly contributed to personalized management for patients with these rare neoplasms. This review aimed to provide an update on the epidemiology, diagnosis, and biomarkers in gastroenteropancreatic neuroendocrine neoplasms.

**Abstract:**

Gastroenteropancreatic neuroendocrine neoplasms (GEP-NENs) are a heterogeneous group of malignancies that originate from the diffuse neuroendocrine cell system of the pancreas and gastrointestinal tract and have increasingly increased in number over the decades. GEP-NENs are roughly classified into well-differentiated neuroendocrine tumors and poorly differentiated neuroendocrine carcinomas; it is essential to understand the pathological classification according to the mitotic count and Ki67 proliferation index. In addition, with the advent of molecular-targeted drugs and somatostatin analogs and advances in endoscopic and surgical treatments, the multidisciplinary treatment of GEP-NENs has made great progress. In the management of GEP-NENs, accurate diagnosis is key for the proper selection among these diversified treatment methods. The evaluation of hormone-producing ability, diagnostic imaging, and histological diagnosis is central. Advances in the study of the genetic landscape have led to deeper understanding of tumor biology; it has also become possible to identify druggable mutations and predict therapeutic effects. Liquid biopsy, based on blood mRNA expression for GEP-NENs, has been developed, and is useful not only for early detection but also for assessing minimal residual disease after surgery and prediction of therapeutic effects. This review outlines the updates and future prospects of the epidemiology, diagnosis, and management of GEP-NENs.

## 1. Introduction

Neuroendocrine neoplasms (NENs) are a group of epithelial tumors with morphological and immunohistochemical features of neuroendocrine differentiation [1]. Recently, the World Health Organization (WHO) published a uniform classification framework for all NENs to resolve the longstanding confusion regarding differences in terminology among organ systems [1]. The disease can arise in most epithelial organs of the body, with the gastrointestinal (GI) tract and pancreas accounting for approximately 50% of the primary sites [2]. Although all NENs share similar configurations and specific neuroendocrine expressions, they behave very differently in relation to the site of origin, histological grade, clinical stage, and hormone production [3,4]. The clinical presentation and prognosis of NENs are diverse; therefore, various diagnostic and therapeutic approaches have been attempted to date. Multidisciplinary management strategies have improved the survival of patients with NENs; however, the prognosis of patients with advanced NENs is still unfavorable [5,6]. Additionally, the etiology of NENs is largely unknown outside of certain hereditary genetic syndromes, such as multiple endocrine neoplasia type 1 syndrome (caused by *MEN1*), MEN 2, von Hippel–Lindau syndrome (*VHL*), and tuberous sclerosis (*TSC1*, *TSC2*) [7]. Recent advances in genomic and epigenetic sciences have provided significant benefits in oncology [8,9,10], whereas the evidence is insufficient for NENs. Although NENs are considered rare, their incidence has been increasing globally, which in turn has received more attention from clinicians and researchers in recent years. This review focuses on the updated findings of the epidemiology, diagnosis, genetic data, and future perspectives of gastroenteropancreatic (GEP)-NENs.

## 2. Epidemiology

The incidence of GEP-NENs increases with age. The median age is 60 years or more in most gastrointestinal (GI)-NENs but reportedly less than 50 years for the appendix and pancreas [11,12]. The incidence is similar among males and females [11,13]. The reported incidence of GEP-NENs has been increasing worldwide [14,15]. A large population-based study using the Surveillance, Epidemiology, and End Results (SEER) database estimated that the age-adjusted incidence of GEP-NENs in 2012 was 3.56 per 100,000 persons in the United States (US) [2]. The incidence has continuously increased over the last four decades, especially in the small intestine, rectum, and pancreas. Increasing trends have also been observed in European countries, where the prevalence of GEP-NENs ranges from 2.1 to 6.6 cases per 100,000 population per recent reports [12,16,17,18,19]. Several population-based studies have been published in Asian countries. In Japan, the age-adjusted incidences of GI-NENs in 2005 and 2016 were 2.10 and 2.84 per 100,000 people, respectively, indicating an approximately 1.3-fold increase, while the incidence of pancreatic NENs in 2005 and 2016 was 1.01 and 0.70 per 100,000 people, respectively, showing a slight decrease [20,21]. In Taiwan, the age-adjusted incidence of GI and pancreatic NENs between 1996 and 2015 had risen from 0.13 to 1.87 cases and from 0.02 to 0.45 cases per 100,000 population, respectively [22]. The most common site was the rectum, comprising 30% of all NENs and 47% of GEP-NENs. A Korean multicenter study reported dramatic changes in the incidence of GEP-NENs, with the incidence in 2009 becoming nine times that reported in 2000. The most significant increase was found in the rectum, while no apparent changes were observed at other sites [23]. The recent age-adjusted incidence of GEP-NENs worldwide is shown in Table 1.

Recent advances in diagnostic techniques, including endoscopy and imaging, are considered to be responsible for the increased prevalence of GEP-NENs, especially for those in the rectum, stomach, and pancreas [11,13,25]. Indeed, the reported incidence of localized and regional NENs has increased more than that of NENs with distant metastases [2,26].

The distribution of GEP-NENs is known to differ regionally [14]. In Asia, rectal NENs are the most prevalent, followed by pancreatic or gastric NENs [20,21,22,23]. In contrast, small intestinal and appendiceal NENs are predominant in Europe (Figure 1) [12,16,17,18].

Although some combinations of biological and environmental backgrounds are considered, the reason for the regional disparities has not been clearly elucidated. Notably, Kessel et al. have reported similar racial disparities in the US; rectal NENs were more likely to occur in Asians and African Americans, but less likely to occur in Whites. In contrast, small intestinal NENs were common in Whites, African Americans, and Hispanics but rare in Asians [27]. This phenomenon suggests that there might be an association between genetic background and the biological characteristics of GEP-NENs.

The behavior of GEP-NENs varies depending on their primary site, grade, and stage [11,28]. For instance, rectal and appendiceal NENs are more likely to be low-grade and localized, with a better prognosis. However, high-grade NENs are common in the pancreas, stomach, and colon. Esophageal NEN, a rare presentation of NEN, is mostly diagnosed in aggressive stages [11,13,21,23,29]. Although improved survival for patients with metastatic GEP-NENs has been reported in the SEER, comparing the period of 2000–2004 with 2009–2012, the overall survival of patients with GEP-NENs with high-grade and distant metastases was still unfavorable [2,6]. While a subset of NENs is functional, presenting with characteristic endocrine-related symptoms, the majority are non-functional [1] and do not present with symptoms until later stages. Therefore, further development in early identification and targeted therapy for GEP-NENs is warranted.

## 3. Diagnosis

The diagnosis of GEP-NENs is based on biopsy, anatomical and functional imaging, and positron emission tomography (PET) with DOTATATE, a gallium (Ga)-68-labeled octreotide derivative, to identify tumors expressing somatostatin receptors (SSTRs). Some blood biomarkers are specific to functional GEP-NENs; however, their utility for comprehensive diagnosis is limited. On the other hand, biomarkers would be helpful for detecting very small tumors, which are difficult to diagnose by imaging or biopsy [30]. We discuss details of biomarkers including novel multianalyte biomarkers developed in recent years in the later Section 5, Biomarkers.

### 3.1. Pathology

The 2019 WHO classification of tumors of the digestive system [1] defines GEP-NENs as G1 (Ki67 < 3%), G2 (Ki67: 3–20%), and G3 (Ki67 > 20%), according to the Ki67 proliferation index. G3 GEP-NENs are classified based on cell morphology and proliferation into well-differentiated G3 and poorly differentiated neuroendocrine carcinomas (NECs). NECs are further morphologically classified into two subtypes: small-cell and large-cell carcinomas. Mixed neuroendocrine–non-neuroendocrine neoplasms (MiNENs) were proposed for mixed tumors with exocrine components (Table 2).

For a proper pathological diagnosis, the morphology, grade, and immunohistochemical staining for chromogranin A (CgA) and synaptophysin should be assessed. SSTR2 is expressed in many NENs, and immunostaining for SSTR2 is useful in assessing tumor differentiation and estimating the effects of somatostatin analog therapy [31,32]. Another promising new immunohistochemical neuroendocrine marker is the transcription factor insulinoma-associated protein 1 (INSM1), which appears to be more specific to neuroendocrine cells than synaptophysin [33].

### 3.2. Endoscopy

Endoscopy with biopsy is the gold standard for diagnosing NENs of the stomach, duodenum, and colorectum [34,35,36]. For NENs in the small intestine, video-capsule endoscopy and double-balloon endoscopy (DBE) are additional endoscopic techniques that are indicated when primary lesions cannot be detected by conventional imaging such as computed tomography (CT), magnetic resonance imaging (MRI), and somatostatin receptor imaging (SRI) [37]. The sensitivity of DBE in identifying the primary lesion in the small intestine was 90% or more, which is considerably higher than those of other imaging modalities [38].

In the diagnosis of pancreatic NENs, endoscopic ultrasound (EUS) can exclude the effects of intestinal gas and subcutaneous fat compared with extracorporeal ultrasound. The sensitivity and specificity of EUS in the diagnosis of pancreatic NENs have been reported to be higher than those of CT [39]. EUS is particularly useful in detecting small pancreatic lesions. EUS-guided fine-needle aspiration (FNA) for cytology and histology was subsequently performed for grading pancreatic NENs, enabling decisions on appropriate treatment strategy. However, it should be noted that the diagnostic rate using EUS-FNA samples might depend on the technical level. It is believed that the low concordance rates for histological grading based on WHO classification between EUS-FNA and resected specimens is due to tumor heterogeneity and the failure of sampling “hot spots” related to a lesion. Therefore, to increase the concordance rate as much as possible, it is important to collect more than 2000 tumor cells from EUS-FNA samples, as recommended by the European Neuroendocrine Tumor Society [40].

### 3.3. CT

CT is a widely used, standardized, and reproducible technique that generally results in high diagnostic yields, making it the basic radiologic diagnostic imaging method for NENs [41]. The sensitivity is 73% for tumors with suspected primary tumors, 95% for unknown primary tumors, 80% for liver metastases, and 75% for extrahepatic metastases. The threshold of detection is 0.5 cm [42]. Morphological imaging may fail to detect small tumors, especially those located in the stomach, duodenum, and small intestine [34].

### 3.4. MRI

MRI is advantageous in the examination of the liver and pancreas and is usually preferred for initial staging and preoperative imaging. Diffusion-weighted MRI is now routinely used in cell-rich tissues, such as tumors, to take advantage of restricted water movement, which facilitates lesion detection. The sensitivity of MRI for detecting pancreatic NETs is 79% (rang 54–100%) [43,44,45]. The sensitivity of MRI in detecting metastatic liver lesions is 91% (range 82–98%), which is superior to that of CT [46,47,48,49,50]. MRI is also advantageous over CT in bone and brain imaging [41].

### 3.5. Functional Imaging

Functional imaging studies are based on the expression of SSTRs by GEP-NETs. Historically, imaging of SSTRs included ^111^indium pentetreotide scintigraphy (Octreoscan^®^); however, ^68^Ga-DOTATATE PET/CT was found to be more accurate and is now the technique of choice [4]. SSTR-PET imaging is regarded as the most sensitive and specific method for detecting NEN and its metastases, with a sensitivity of 93–96% and specificity of 85–100% [51]. ^68^Ga-DOTATATE PET/CT is also important for determining radionuclide uptake, which is associated with response to peptide receptor radionuclide therapy (PRRT) [52]. PET/CT is often performed for the imaging of GEP-NENs, but since MRI provides greater contrast in soft tissues than CT, PET/MRI is more appropriate, particularly when liver and bone metastases are suspected and need to be excluded [53]. Other methods such as ^64^Cu-DOTATATE are currently in use. It has been reported that ^64^Cu-DOTATATE has a higher detection rate than ^68^Ga-DOTATATE. In the future, ^68^Ga-DOTATATE might be replaced by ^64^Cu-DOTATATE [54,55,56].

## 4. Genetic Features and Targeted Therapy

Data regarding somatic mutations in GEP-NENs were obtained and analyzed for a total of 859 specimens collected from 820 patients from the AACR Project GENIE database (ver.10) (https://www.aacr.org/professionals/research/aacr-project-genie/ accessed on 20 August 2021) (Figure 2, Appendix A). A total of 490, 191, 36, 19, and 124 specimens from 464, 182, 35, 18, and 121 patients with pancreatic NETs (PANET), small bowel well-differentiated NETs (SBWDNET), well-differentiated NETs of the rectum (RWDNET), well-differentiated NETs of the appendix (AWDNET), and high-grade NECs of the colon and rectum (HGNEC) were included, respectively. These NETs include grade 1 to grade 3 specimens, and NECs include both small and large types of poorly differentiated neuroendocrine carcinoma of the WHO classification.

PANET mainly harbors mutations in genes that encode regulators of the PI3K/mTOR pathway. The most frequently mutated gene was *MEN1,* with variants detected in 30.2% of the patients, followed by 14.9% in *DAXX*, 14.5% in *TP53*, 7.8% in *ATRX*, and 6.9% in *TSC2*. *MEN1*, *DAXX*, and *ATRX* genes play important roles in chromatin remodeling. MEN1 binds to the *TERT* promoter and affects the machinery that controls telomere integrity [57]. Inactivating mutations in *DAXX* and *ATRX* are strongly correlated with somatic telomere repeat content and telomere length [58]. Mutations in *DAXX*, *ATRX*, and *MEN1* are associated with a worse prognosis than the corresponding genes without these mutations [59,60]. These mutations are rarely present in gastrointestinal NETs. *TP53* mutations were predominantly found in poorly differentiated pancreatic NECs and G3 PANET [58,61,62], with mutations detected in 62.9% of HGNEC (Figure 2).

SBWDNET and RWDNET have a low rate of candidate driver events. *CDKN1B* mutations were most frequently identified in SBWDNET, as previously reported [63]. *ERBB2* mutations were frequently identified in RWDNET, and others have found recurrent mutations in *TP53*, *PTEN*, and *SMAD4*, as in a previous report [64]. In well-differentiated neuroendocrine tumors of the appendix, mutations in *KRAS* and *PIK3CA* genes, which frequently occur in right-sided colorectal cancer [65], were detected (Figure 2).

In HGNEC, high mutation rates of colorectal adenocarcinoma-associated genes such as *APC*, *KRAS*, *BRAF*, and *TP53* were found. *BRAF* mutations were detected in 16.1% of patients with HGNEC. *BRAF* mutations occur in 5–10% of patients with advanced colorectal adenocarcinomas and are associated with a poor prognosis [66,67]. Furthermore, HGNEC displays a high frequency of recurrent *TP53* and *RB1* mutations, which are commonly observed in small-cell lung cancer [68], and are rare events in other NETs. These mutations may play critical roles in the aggressiveness of malignant tumors (Figure 2).

All types of GEP-NENs had at least one potential actionable mutation that was predictive of a drug response according to the evidence levels of 1–3B in OncoKB (http://oncokb.org accessed on 30 August 2021) (Figure 3) [69].

For instance, 40 patients (8.1%) with pancreatic NETs may have benefited from mTOR inhibitors. The RADIANT-3 clinical trial with the mTOR inhibitor everolimus demonstrated its safety and efficacy in the treatment of advanced PANET [70]. Moreover, a phase 2 pilot study is currently investigating the utility of the mTOR inhibitor ABI-009 as a single agent in patients with metastatic, unresectable, low- or intermediate-grade NETs of the lung or GEP system (NCT03670030). Eighteen patients (14.5%) with HGNEC harbored a *BRAF* V600E mutation, which was inhibited by vemurafenib. A previous report demonstrated vemurafenib responses in two patients with NECs [71]; one had a partial response that was sustained for 4.1 months, and the other had stable disease (SD) of unknown duration. The data for utilizing candidate genes for patients with GEP-NENs are insufficient, and future studies need to identify a novel therapeutic target.

## 5. Biomarkers

There are no established biomarkers for patients with GEP-NENs. If patients have symptoms suspected for functional GEP-NENs, some biomarkers, such as insulin, gastrin, and glucagon, are specific, although their use is limited in accurate diagnosis [30]. Patients with functional NENs could benefit from somatostatin analogs to relieve their hormonal symptoms [3,4]. Additionally, hereditary endocrine tumor syndromes, including MEN1 and VHL, might present in the background of these patients; therefore, attention to multifocal and multiorgan tumors is needed [7]. Since CgA has been commonly used as a blood-based biomarker for NET, regardless of tumor types (functionality or location), its accuracy has been discussed in recent studies [30,72,73]. Several factors such as heart failure, renal failure, malignant tumors, and the use of medication with proton-pump inhibitors may cause false-positive CgA results [30,72].

In recent years, the analysis of somatic mutations associated with NETs has provided a new strategy for their diagnosis or follow-up. Liquid biopsy, based on mRNA, is thought to be useful as a novel biomarker for NENs instead of monoanalyte biomarkers. The analysis of the NET transcriptome signature, NETest (Wren Laboratories, Branford, CT, USA), is accurate as a circulating multianalyte biomarker [73]. NETest is a prespotted polymerase chain reaction (PCR) plate targeting 51 genes, in which tumor-derived mRNA is extracted from the patient’s blood and quantified by PCR [30,74]. The output results show 0–100% as an activity index, and the cut-off value is 20%. An index of 20–40% is considered an SD and 41–100% a progressive disease (PD) [30]. NETest shows high sensitivity and specificity for diagnosis (Table 3). The diagnostic accuracy of NETest is significantly higher (99%) than that of CgA (21–36%) for GEP-NENs [72].

NETest is especially valuable in terms of follow-up after radical resection of NETs. After R0 resection, the NETest index significantly decreased from 62% to 22% 30 days after the initial surgery. For 30% of patients who underwent R0 resection, the NETest index remained high (≥20%); 81% of those patients experienced recurrence 18 months after the initial surgery [79]. The high NETest index after tumor resection suggests the existence of minimal residual disease (MRD) and early recurrence [79,80].

PRRT is thought to be an effective therapeutic option for unresectable or relapsed NETs. PRRT using 177Lu-DoTATATE was approved by the US Food and Drug Administration (FDA) in 2018. Among “Responder” patients after PRRT, NETest score significantly decreased from 61% to 29%, while “Non-responders” showed unchanging or increasing scores [81].

NETest is also adequate for the evaluation of disease progression and prognosis. In total, 87% of patients diagnosed with SD by the RECIST1.1 had a low NETest score (≤40%), whereas 81% of patients with PD showed a high NETest score (≥80%). Comparison of the three classes of NETest scores (low: <40%; intermediate: 41–79%; and high biological activity: 80–100%) indicates shortening of progression-free survival in the intermediate and high-biological-activity groups [80]. NETest reflects disease activity, and a high score indicates a poor response to drug therapies or PRRT. The multianalyte biomarker, NETest, has multiple uses. It is used not only for the diagnosis of GEP-NENs but also for the determination of disease activity and therapy effectiveness and follow-up after tumor resection. NETest can detect disease progression 5–24 months before imaging changes. Identification of MRD that cannot be detected by imaging studies should lead to earlier therapeutic intervention in GEP-NENs [76,78]. Follow-up of GEP-NENs requires frequent endoscopy with biopsy and/or CT scanning, which causes physical pain and radiation exposure, and is costly. In the US, using a follow-up strategy with NETest resulted in a 42% saving in cost [82]. NETest would also be effective in reducing these patients’ burdens.

GEP-NENs are highly heterogeneous diseases, which complicates their diagnosis or evaluation of progression. Although NETest shows highly sensitive and specific results as presented above, a comprehensive genetic analysis of GEP-NENs is needed for more accurate diagnosis and early therapeutic intervention in the future.

## 6. Conclusions

There has been a rapid increase in the number of clinically identified GEP-NENs in the last few decades. Given the different distribution of GEP-NENs among races, there might be a biological difference based on genetic background; hence, evidence from the Asian population is required. Recently, next-generation sequencing has provided new insights into the genetic and epigenetic landscape of a subset of GEP-NENs [5]. Various therapeutic options are currently available for treating GEP-NENs. Although surgery is the first choice for resectable GEP-NENs, drug therapies, such as somatostatin analogs, molecular-targeted drugs, and cytotoxic agents, play a key role in the treatment of unresectable or relapsed GEP-NENs [83]. With regard to molecular-targeted drugs, sunitinib is available for pancreatic NETs, whereas everolimus is used for all types of NETs. Recently, Japan approved the use of the agent in PRRT, which had already been approved by the FDA and was broadly used for the treatment of GEP NETs in Europe and in the US [72,83]. The efficacy of immune checkpoint inhibitors for GEP-NENs remains controversial; meanwhile, some clinical trials are ongoing [83,84]. Although these novel and personalized therapeutic options are expected to improve the prognosis of patients with GEP-NENs, their application in clinical settings is still limited. To fill this gap, the development of optimized diagnostic modules and therapies is underway. For instance, constant molecular monitoring via liquid biopsy might be a predictive tool for tailoring a personalized diagnostic and treatment strategy that improves patient outcomes. It will require an international and transdisciplinary endeavor to enter all patients with these uncommon neoplasms into a novel personalized clinical trial.

## Figures and Tables

**Figure 1 cancers-14-01119-f001:**
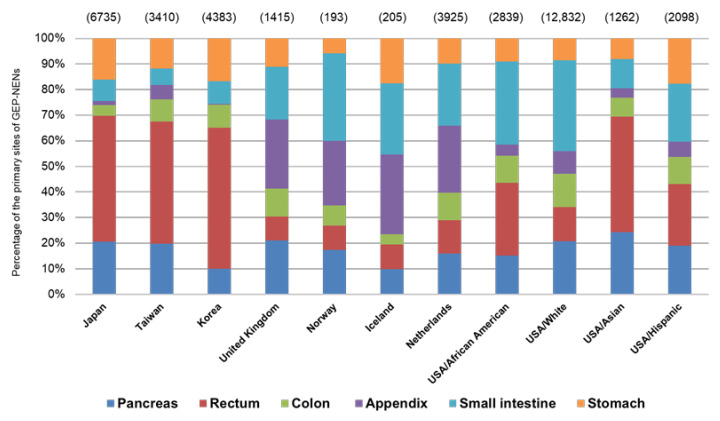
Distribution of the primary sites of NENs. Numbers shown in parentheses denote the study samples in each reference. USA: United States of America. Data for the figure is based on references [12,16,17,18,21,22,23,27].

**Figure 2 cancers-14-01119-f002:**
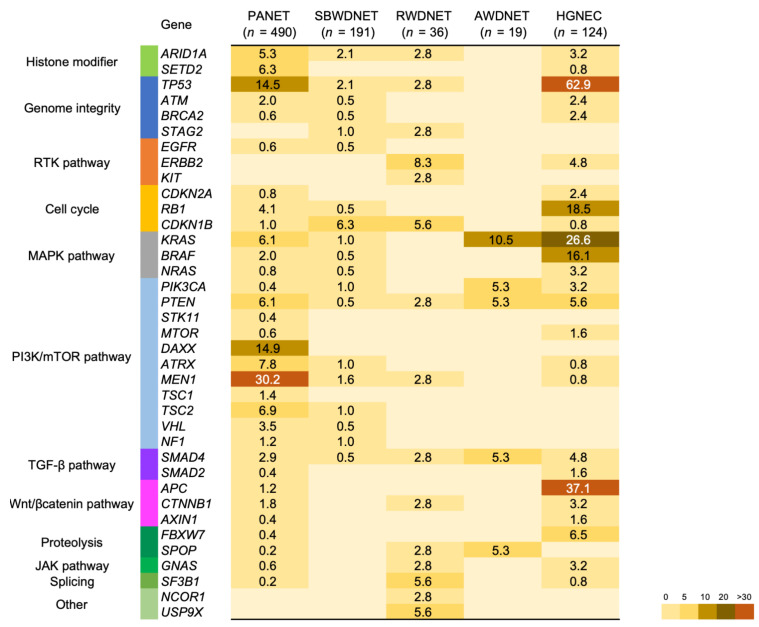
Mutation frequencies in neuroendocrine neoplasms arising from the gastrointestinal tract and pancreas. The percentages of samples mutated in individual tumor types are shown. PANET: Pancreatic neuroendocrine tumor, SBWDNET: Small bowel well-differentiated neuroendocrine tumor, RWDNET: Well-differentiated neuroendocrine tumor of the rectum, AWDNET: Well-differentiated neuroendocrine tumor of the appendix, HGNEC: High-grade neuroendocrine carcinoma of the colon and rectum.

**Figure 3 cancers-14-01119-f003:**
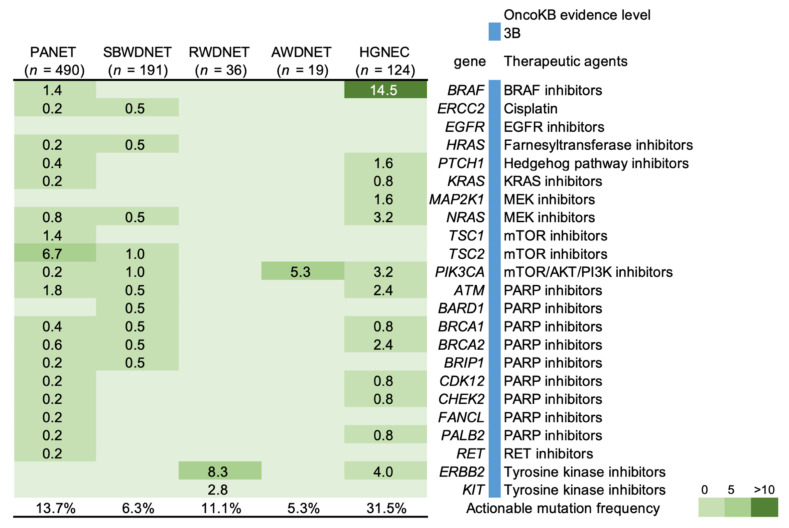
Frequency of actionable genetic mutations in gastroenteropancreatic neuroendocrine neoplasms. Percentages of samples mutated in individual tumor types are shown. PANET: Pancreatic neuroendocrine tumor, SBWDNET: Small bowel well-differentiated neuroendocrine tumor, RWDNET: Well-differentiated neuroendocrine tumor of the rectum, AWDNET: Well-differentiated neuroendocrine tumor of the appendix, HGNEC: High-grade neuroendocrine carcinoma of the colon and rectum.

**Table 1 cancers-14-01119-t001:** Age-adjusted incidence of GEP-NENs, according to country.

Country	Reference	GEP-NEN Incidence (Cases Per 100,000) *	Data Time Period
Netherlands	[12]	2.12	2001–2010
Germany	[19]	2.2	2006
Taiwan	[22]	2.31	2015
Japan	[21]	3.53	2016
United States of America	[2]	3.56	2012
Iceland	[17]	3.85	2000–2014
Australia	[24]	4.46	2006–2015
United Kingdom	[16]	4.6	2015
Norway	[18]	6.62	2009

* In order of incidence.

**Table 2 cancers-14-01119-t002:** World Health Organization (WHO) 2019 classification for neuroendocrine neoplasms of the gastrointestinal tract and hepatopancreatobiliary organs. The table is modified from [1].

Definition	Cell Morphology	Ki67 Proliferative Index ^a^	Mitotic Count ^b^
NET G1	Well-differentiated	<3%	<2
NET G2		3–20%	2–20
NET G3		>20%	>20
NEC	Poorly differentiated	>20%	>20
Small-cell type			
Large-cell type			
MiNEN	Well- or poorly differentiated	Variable	Variable

NET, neuroendocrine tumor; NEC, neuroendocrine carcinoma; MiNEN, mixed neuroendocrine–non-neuroendocrine neoplasm. ^a^ Ki67 proliferative index is determined by counting ≥500 cells in the regions of highest labeling; ^b^ Mitotic rates are expressed as the number of mitoses/2 mm^2^ determined in 50 fields of 0.2 mm^2^; the final grade is based on the proliferation index that places the neoplasm in the higher-grade category.

**Table 3 cancers-14-01119-t003:** Diagnostic sensitivity, specificity, and accuracy of NETest.

Author	Sites of NET	Sensitivity (%)	Specificity (%)	Accuracy (%)
van Treijen et al., 2018 [75]	GEP	89	72	nd
Malczewska et al., 2019 [76]	P, SI	99	95	97
Liu et al., 2019 [77]	GEP, BP, U	nd	nd	96
Malczewska et al., 2020 [78]	G	100	87	90

GEP: Gastroenteropancreatic, P: Pancreatic, SI: Small intestine, BP: Bronchopulmonary, U: unknown primary, G: Gastric, nd: no data.

## Data Availability

The data presented in this study are available in the present manuscript or in the Appendix A.

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
