# Peer review of "Update on Epidemiology, Diagnosis, and Biomarkers in Gastroenteropancreatic Neuroendocrine Neoplasms"

_cancers, 2022, doi:10.3390/cancers14051119_

Round 1

Reviewer 1 Report

The authors should be commended on their effort to take this encompassing topic. However, there are significant issues with regards to the content of the article and its style. 

With regards to content:

  1. Though the article is titled "Update on the management strategy for gastroenteropancreatic neuroendocrine neoplasms," there is minimal content on management of advanced disease (surgical debulking, liver directed therapy, novel systemic therapies for NETs, more discussion of PRRT and PRRT developments in NETs and therapeutic approaches for NECs). The article is much more focused on epidemiology, diagnosis and biomarkers.
  2. In simple summary, the increasing incidence of NENs is not just due to improved diagnostic imaging. There are also as of yet undetermined biologic factors contributing to this. 
  3. Under diagnosis, only need to focus on functional imaging advances (SRI) and pathologic grading advances.
  4. Under biomarkers, would say there are no established biomarkers. There are multiple promising biomarkers such as the NETest and ctDNA.
  5. Under targeted therapies, there are no biomarkers in NENs outside of MSI, NTRK and possibly BRAFV600E which are truly actionable at this time. To say otherwise is premature. 

With regards to style:

  1. There are multiple instances in which patient focused language should be used (e.g. patients with advanced NENs)

Author Response

Reviewer 1

The authors should be commended on their effort to take this encompassing topic. However, there are significant issues with regards to the content of the article and its style.

With regards to content:

  1. Though the article is titled "Update on the management strategy for gastroenteropancreatic neuroendocrine neoplasms," there is minimal content on management of advanced disease (surgical debulking, liver directed therapy, novel systemic therapies for NETs, more discussion of PRRT and PRRT developments in NETs and therapeutic approaches for NECs). The article is much more focused on epidemiology, diagnosis and biomarkers.

We appreciate this important comment raised by the reviewer. Accordingly, we have changed the title to “Update on epidemiology, diagnosis, and biomarkers in gastroenteropancreatic neuroendocrine neoplasms” to better represent our review.

  1. In simple summary, the increasing incidence of NENs is not just due to improved diagnostic imaging. There are also as of yet undetermined biologic factors contributing to this.

We agree with the point that the increasing incidence of GEP-NENs is not solely due to improved diagnostic imaging. Wehave revised the related part in the Simple Summary as below (page 1, lines 20–23).

Simple Summary: Neuroendocrine neoplasms are divided into two groups: well-differentiated neuroendocrine tumors and poorly differentiated neuroendocrine carcinomas. The progress in diagnostic methods, including pathology optimization and imaging, might be one of the reasons for the increasing incidence of gastroenteropancreatic neuroendocrine neoplasms; however, remaining biological factors are undetermined. Rapid advances in molecular diagnostic and treatment strategies in recent years have significantly contributed to personalized management for patients with these rare neoplasms. This review aimed to provide an update on the epidemiology, diagnosis, and biomarkers in gastroenteropancreatic neuroendocrine neoplasms.

  1. Under diagnosis, only need to focus on functional imaging advances (SRI) and pathologic grading advances.

Thank you for your important suggestion. As the reviewer indicated, we have mainly focused on the advances infunctional imaging (SRI) and histological grading (page 4, 3.2. Endoscopy). Although there have been fewer updates on endoscopy, CT, and MRI compared with those in SRI and pathological diagnosis, we believe that recent knowledge on these modalities is also important for readers and should be included.

  1. Under biomarkers, would say there are no established biomarkers. There are multiple promising biomarkers such as the NETest and ctDNA.

We appreciate this important comment raised by the reviewer. We agree with you and have replaced the term “promising” with ”established” in the text as follows (page 8, line 269):

There are no established biomarkers for patients with GEP-NENs.

  1. Under targeted therapies, there are no biomarkers in NENs outside of MSI, NTRK and possibly BRAFV600E which are truly actionable at this time. To say otherwise is premature.

We appreciate this critical comment raised by the reviewer. We completely agree with you. We have added a sentence to describe this issue in the text as follows (page 8, lines 266–267):

The data for utilizing candidate genes for patients with GEP-NENs are insufficient, and future studies need to identify a novel therapeutic target.

With regards to style:

  1. There are multiple instances in which patient focused language should be used (e.g. patients with advanced NENs)

As the reviewer pointed out, there were several sentences where patient-focused language should have been used. We have revised the related parts in the text:

Page 2, line 58-59: patients with NENs

Page 3, lines 118: patients with metastatic GEP-NENs

Reviewer 2 Report

This is an extensive and concise reviews of GEP-NENs.

Please check the following points

  1. In the text, it is stated that the incidence rate has been increasing in recent years, so it could be in chronological order for Table 1. I think it would be easier to understand if the title or description of the table says that it is in the order of incidence rate.
  2. The USA in Figure 1 is a quote from Paper 27, which I checked it, but since the text says, "African Americans," I think it should be AA or something similar instead of Black.
  3. Table 2 has clearly been modified from the original work and should be marked as such.
  4. On page 4, line 154, US is seeming to stand for ultrasound or ultrasonography, but since US is used as an abbreviation for united states, it is more helpful to spell it out.
  5. It is difficult to comment on Figure 2 and 3 as I cannot refer to the original data, but I think the term “High grade” neuroendocrine carcinoma needs explanation, especially as it is not consistent with the WHO classification.

Author Response

Reviewer2

This is an extensive and concise reviews of GEP-NENs.

Please check the following points

  1. In the text, it is stated that the incidence rate has been increasing in recent years, so it could be in chronological order for Table 1. I think it would be easier to understand if the title or description of the table says that it is in the order of incidence rate.

Thank you for the kind notice. We have added a footnote to Table 1 to explain that the data are “in order of incidence” (page 3, line 92).

  1. The USA in Figure 1 is a quote from Paper 27, which I checked it, but since the text says, "African Americans," I think it should be AA or something similar instead of Black.

According to the reviewer’s suggestion, we have changed the term “Black” to “African American” in Figure 1, consistent with the text.

  1. Table 2 has clearly been modified from the original work and should be marked as such.

We agree with the reviewer’s comment and have added a footnote to explain that this table is modified from “Reference [1]” as follows (page 4, line 140–141):

Table 2. World Health Organization (WHO) 2019 classification for neuroendocrine neoplasms of the gastrointestinal tract and hepatopancreatobiliary organs. The table is modified from [1].

  1. On page 4, line 154, US is seeming to stand for ultrasound or ultrasonography, but since US is used as an abbreviation for united states, it is more helpful to spell it out.

According to the reviewer’s comment, we have changed the related part in the text as follows (page 5, lines 163–164):

In the diagnosis of pancreatic NENs, endoscopic ultrasound (EUS) can exclude the effects of intestinal gas and subcutaneous fat compared with extracorporeal ultrasound.

  1. It is difficult to comment on Figure 2 and 3 as I cannot refer to the original data, but I think the term “High grade” neuroendocrine carcinoma needs explanation, especially as it is not consistent with the WHO classification.

We appreciate this important comment raised by the reviewer. We have added a description of this in the Genetic features and targeted therapy section as follows (page 6, lines 215–217):

These NETs include from grade 1 to grade 3 specimens, and NECs include both small and large types of poorly differentiated neuroendocrine carcinoma of the WHO classification.

Reviewer 3 Report

Overall, I enjoyed reading the paper. Flows well but does have some major drawback as related to the title.

  1. The title is: "Management Strategy"....but unfortunately NO strategy is discussed. A review paper needs to be comprehensive and comparative. Their management strategy needs to separate Well Differentiated from Poorly Differentiated NECs. What is the role of surgery? What is the role of Somatostatin therapy? What is the role of Chemotherapy? Is there a role for IR for the management of Liver Mets? Role of Radioembolization Vs TACE?What is the role of PRRT? When do you transition from one therapy to the other and what are the results of those therapies.
  2. Line 119: Please elaborate on what biomarkers are drawn at the initial consultation and then how this impacts their treatment plan?
  3. Line 160: Please elaborate on EUS guided core biopsies of NETs. Is there a role for lymphovascular invasion on the core biopsy and how that might influence treatment.
  4. Line 184: Elaborate on 64Cu- DOTATATE. Whatis the benefit? Is it likely to replace 68Ga- PET scan?
  5. Any role for PET-MRI's in NETs?
  6. Figure 2: This figure includes pancreas, small bowel, rectum, appendix....What does GINET in column 2 refer to?

Author Response

Reviewer3

Overall, I enjoyed reading the paper. Flows well but does have some major drawback as related to the title.

  1. The title is: "Management Strategy"....but unfortunately NO strategy is discussed. A review paper needs to be comprehensive and comparative. Their management strategy needs to separate Well Differentiated from Poorly Differentiated NECs. What is the role of surgery? What is the role of Somatostatin therapy? What is the role of Chemotherapy? Is there a role for IR for the management of Liver Mets? Role of Radioembolization Vs TACE? What is the role of PRRT? When do you transition from one therapy to the other and what are the results of those therapies.

Thank you for the critical comment. We agree that there has been substantial progress in the local and systemic treatment of GEP-NENs, including surgery, chemotherapy, and radiation. Meanwhile, biological background and diagnosis are also important to develop a management plan, so we have focused on these factors in this review. However, as the reviewer pointed out, the title of our manuscript might be misleading for the readers. Therefore, we have changed the title to “Update on epidemiology, diagnosis, and biomarkers in gastroenteropancreatic neuroendocrine neoplasms.”

Although it is not practical to cover all topics on the treatment in this article, we have mentioned diagnostic modalities and related treatments in each section. In particular, somatostatin analogs (page 4, lines 148–150) and PRRT (page 5, lines 198–200) have been used for NENs expressing somatostatin receptors. Analyzing the mutation profile of NENs would be informative in optimizing chemotherapy and molecular targeted therapy (pages 6–8, section 4. Genetic features and targeted therapy). In addition, there have been new biomarkers expected to predict the response to drug therapies and PRRT (page 9, lines 300–304).

  1. Line 119: Please elaborate on what biomarkers are drawn at the initial consultation and then how this impacts their treatment plan?

We appreciate this important comment raised by the reviewer. At the initial consultation of patients, we would check some blood-based biomarkers. Chromogranin A is commonly used regardless of tumor type. If patients have symptoms suspecting functional GEP-NENs, hormones such as insulin, gastrin, and glucagon are useful. These monoanalyte biomarkers are helpful to determine the presence, functionality, and location of the tumor; however, they are limited in their accuracy. On the other hand, it is difficult to detect very small tumors even though imaging and histology are important and decisive strategies for diagnosis. The combination of blood biomarkers and imaging/histology is needed for a more accurate and early diagnosis of GEP-NENs.

To address the reviewer’s comment, we have added and modified the description in the text (section 3. Diagnosis and section 5. Biomarkers).

Page 4, lines 124–131
The diagnosis of GEP-NENs is based on biopsy, anatomical and functional positron emission tomography (PET) with DOTATATE, a gallium (Ga)-68-labeled octreotide derivative, to identify tumors expressing somatostatin receptors (SSTRs). Some blood biomarkers are specific to functional GEP-NENs; however, their utility for comprehensive diagnosis is limited. On the other hand, biomarkers would be helpful for detecting very small tumors, which are difficult to diagnose by imaging or biopsy [30]. We discuss details of biomarkers including novel multianalyte biomarkers developed in recent years in a later chapter (5. Biomarkers).

Page 8, line 269–277

There are no established biomarkers for patients with GEP-NENs. If patients have symptoms suspected for functional GEP-NENs, some biomarkers, such as insulin, gastrin, and glucagon, are specific, although their use is limited in accurate diagnosis [30]. Patients with functional NENs could benefit from somatostatin analogs to relieve their hormonal symptoms [3,4]. Additionally, hereditary endocrine tumor syndromes, including MEN1 and VHL, might present in the background of these patients; therefore, attention to multifocal and multiorgan tumors is needed [7]. Since CgA has been commonly used as a blood-based biomarker for NET, regardless of tumor types (functionality or location), its accuracy has been discussed in recent studies [30, 72, 73].

  1. Line 160: Please elaborate on EUS guided core biopsies of NETs. Is there a role for lymphovascular invasion on the core biopsy and how that might influence treatment.

Thank you for providing these insights. First, the presence of lymphovascular invasion is not diagnosed with EUS-FNA (fine-needle biopsy). However, the biopsy under EUS-FNA is helpful for pretreatment diagnosis or predicting the histological grade of NENs. We have noted this point in the text (page 5, lines 166–169). In addition, we have added a description (page 5, lines 170–175) to explain the technical tips and limitations on EUS-FNA as follows:

It is believed that the low concordance rates for histological grading based on WHO classification between EUS-FNA and resected specimens is due to tumor heterogeneity and the failure of sampling “hot spots” related to a lesion. Therefore, to increase the concordance rate as much as possible, it is important to collect more than 2000 tumor cells from EUS-FNA samples, as recommended by the European Neuroendocrine Tumor Society [40].

  1. Line 184: Elaborate on 64Cu- DOTATATE. Whatis the benefit? Is it likely to replace 68Ga- PET scan?

Thank you for your important suggestion. 64Cu-DOTATATE is an alternative isotope characterized by a long half-life (12.7 h for 68 min of 68Ga) and a short positron range (2.9 mm for 0.6 mm). Since 64Cu-DOTATATE can be imaged 3 hafter injection, it matches the receptor uptake kinetics, and the target-background ratio and resolution are high, so the detection rate is higher than that of 68Ga-DOTATATE. It is considered that 64Cu-DOTATATE might be replaced by 68Ga-DOTATATE in the future. We have added a description on 64Cu- DOTATATE in the text (page 5, lines 203–205).

It has been reported that 64Cu-DOTATATE has a higher detection rate than 68Ga-DOTATATE. In the future, 64Cu-DOTATATE might be replaced by 68Ga-DOTATATE [54-56].

  1. Any role for PET-MRI's in NETs?

We appreciate this important comment raised by the reviewer. It has been reported that PET/MRI shows advantages in the detection of liver and bone metastases of NENs compared to PET/CT. Accordingly, we added a description as follows (page 5, line 200–202):

PET/CT is often performed for the imaging of GEP-NENs, but since MRI provides greater contrast in soft tissues than CT, PET/MRI is more appropriate, particularly when liver and bone metastases are suspected and need to be excluded [53]

  1. Figure 2: This figure includes pancreas, small bowel, rectum, appendix....What does GINET in column 2 refer to?

We appreciate this valuable comment raised by the reviewer. Since GINETs are NETs that occur in the gastrointestinal tract but has an unknown organ, it is difficult to interpret the data; thus, we decided to exclude GINETs from the analysis. We have revised the related part in the manuscript (page 6, lines 211–217, and Figure 2):

A total of 490, 191, 36, 19, and 124 specimens from 464, 182, 35, 18, and 121 patients with pancreatic NETs (PANET), small bowel well-differentiated NETs (SBWDNET), well-differentiated NETs of the rectum (RWDNET), well-differentiated NETs of the appendix (AWDNET), and high-grade NECs of the colon and rectum (HGNEC) were included, respectively. These NETs include from grade 1 to grade 3 specimens, and NECs include both small and large types of poorly differentiated neuroendocrine carcinoma of the WHO classification.

Round 2

Reviewer 1 Report

The authors have done a commendable job revising the manuscript. I just a few minor additions:

Line 204- I believe authors were trying to say the opposite. Copper-64 dotatate may replace gallium dotatate not vice versa.

Line 306- SD by RECIST 1.1 not in RECIST 1.1

Line 336 and 337 -treatment of gastroenteropancreatic NETs (not just midgut NETs)

Author Response

The authors have done a commendable job revising the manuscript. I just a few minor additions:

  1. Line 204- I believe authors were trying to say the opposite. Copper-64 dotatate may replace gallium dotatate not vice versa.

We appreciate this important oversight raised by the Reviewer. We have changed the indicated part of the text as follows.

Page 5, lines 204-205: In the future, 68Ga-DOTATATE might be replaced by 64Cu-DOTATATE [54-56].

  1. Line 306- SD by RECIST 1.1 not in RECIST 1.1

We have changed the text accordingly.

Page 9, lines 305–306: Eighty-seven percent of patients diagnosed with SD by the RECIST1.1 had a low NETest score (≤40%), whereas 81% of patients with PD showed a high NETest score (≥80%).

  1. Line 336 and 337 -treatment of gastroenteropancreatic NETs (not just midgut NETs)

We have changed the indicated  text accordingly.

Page 9, lines 334-337: Recently, Japan approved the use of the agent PRRT, which had already been approved by the FDA and was broadly used for the treatment of GEP NETs in Europe and in the US [72,83].
